# Evaluation of an organisational-level monetary incentive to promote the health and wellbeing of workers in small and medium-sized enterprises: A mixed-methods cluster randomised controlled trial

Lena Al-Khudairy[1]*, Yasmin Akram[2], Samuel I. Watson[3], Laura Kudrna[3], Joanna Hofman[4], Madeline Nightingale[4], Lailah Alidu[5], Andrew Rudge[2], Clare Rawdin[1], Iman Ghosh[1], Frances Mason[1], Chinthana Perera[1], Jane Wright[1], Joseph Boachie[1], Karla Hemming[2], Ivo Vlaev[6], Sean Russell[2], Richard J. Lilford[3]

1 Warwick Medical School, University of Warwick, Coventry, United Kingdom, 2 West Midlands Combined Authority, Birmingham, United Kingdom, 3 Institue Applied Health Research, University of Birmingham, Edgbaston, United Kingdom, 4 RAND Europe, Cambridge, United Kingdom, 5 Kingston University, Surray, United Kingdom, 6 Warwick Business School, University of Warwick, Coventry, United Kingdom

* Lena.al-khudairy@warwick.ac.uk

**Data Availability Statement:** Type of analyses for which data will be made available: for specified

## Abstract

We conducted an independent evaluation on the effectiveness of an organisational-level monetary incentive to encourage small and medium-sized enterprises (SMEs) to improve employees' health and wellbeing. This was A mixed-methods cluster randomised trial with four arms: high monetary incentive, low monetary incentive, and two no monetary incentive controls (with or without baseline measurements to examine 'reactivity' The consequence of particpant awareness of being studied, and potential impact on participant behavior effects). SMEs with 10–250 staff based in West Midlands, England were eligible. We randomly selected up to 15 employees at baseline and 11 months post-intervention. We elicited employee perceptions of employers' actions to improve health and wellbeing; and employees' self-reported health behaviours and wellbeing. We also interviewed employers and obtained qualitative data. One hundred and fifty-two SMEs were recruited. Baseline assessments were conducted in 85 SMEs in three arms, and endline assessments in 100 SMEs across all four arms. The percentage of employees perceiving "positive action" by their employer increased after intervention (5 percentage points, pp [95% Credible Interval -3, 21] and 3pp [–9, 17], in models for high and low incentive groups). Across six secondary questions about specific issues the results were strongly and consistently positive, especially for the high incentive. This was consistent with qualitative data and quantitative employer interviews. However, there was no evidence of any impact on employee health behaviour or wellbeing outcomes, nor evidence of 'reactivity'. An organisational intervention (a monetary incentive) changed employee perceptions of employer behaviour but did not translate into changes in employees' self-reports of their own health behaviours or wellbeing.

purpose Mechanism by which data will be made available: approval of a proposal and a signed data access agreement Length of availability will be informed by the submitted proposal. Our ethical consideration under Data management allows us to keep participant data for 10 years once created according to the University of Warwick's guidelines. Our ethics committee approved data sharing with our collaborators RAND Europe and West Midlands Combined Authority. For any future requests we would have to contact our ethics committee referencing our ethical approval letter. We contacted the ethics committee and they stated the following: The challenge here seems to be that the BSREC ethics committee and participants were informed that data would only be shared with collaborators. Given this, I am happy for you to list BSREC as the contact point for this (bsrec@warwick.ac.uk). If they need an individual, feel free to provide my details – Dale.Topley@Warwick.ac.uk.

**Funding:** This work was supported by the Work and Health Unit for LA-K, YA, SIW, LK, JH, MN, LA, AR, CR, IG, FM, JW, JB, IV, SR, and RL and by the National Institute of Health Research – Applied Research Collaboration West Midlands for LA-K, SIW, LK, LA, CR, IG, CP, KH, and RL. The Work and Health Unit agreed the protocol but had no role in data collection and analysis, decision to publish, or preparation of the manuscript. The National Institute of Health Research – Applied Research Collaboration West Midlands had no role in study design, data collection and analysis, decision to publish, or preparation of the manuscript.

**Competing interests:** The authors have declared that no competing interests exist.

**Trial registration:** AEARCTR-0003420, registration date: 17.10.2018, retrospectively registered (delays in contracts and identfying a suitable trial registry). The authors confirm that there are no ongoing and related trials for this intervention.

## Introduction

Given the large role that work plays in peoples' lives and the causal association between behaviour and ill health, an argument can be made that employers should encourage their workforce to adopt healthy behaviours [1, 2]. One way to encourage employers to take action to promote the health and wellbeing of their workforce is to reward them for doing so using monetary incentives. The use of monetary rewards to encourage organisations to promote employee health has been widely adopted, being embedded in the Affordable Care Act in the US, and in the Commissioning Quality and Innovation (CQUIN) policy and Wellbeing Challenge Fund in the UK [1, 3]. However, in keeping with a general lack of rigorous evaluations of workplace wellness initiatives [4], the use of monetary incentives to encourage organisations to promote employee health has not been evaluated in randomised trials.

The lack of rigorous evaluation is not due to lack of interest. A recent overview identified more than 400 literature reviews of workplace initiatives to promote health and wellbeing [5]. We examined over 900 studies identified by three of these reviews [6–8], locating only five studies that isolated the effect of organisational-level monetary incentives [9–13]. None of the five studies was a randomised controlled trial, and all were about paying organisations for their performance (P4P). P4P is an area of research largely about the quality of healthcare provided to patients rather than the health and wellbeing of healthcare staff. Nevertheless, staff outcomes were included in three of these studies, showing P4P associated with better motivation [10], staff satisfaction [11], and staff perceptions of working conditions [13].

Here we report a study that examines the effects of monetary incentives using a causal approach. The incentive was designed and funded by the Work and Health Unit (WHU) of the UK government. It was implemented by staff from the West Midlands Combined Authority's (WMCA) 'Local Government Implementation Team'. Before implementing the incentive, the WHU commissioned us to conduct a prospective evaluation. We proposed and implemented a cluster randomised design with multiple quantitative and qualitative outcomes, which included employee perceptions of employers' actions to improve health and wellbeing (the employer 'offer'), and employees' self-reported health behaviours and wellbeing. We also interviewed employers and obtained qualitative data.

## Methods

### Ethics statement

The study was conducted in the West Midlands region of England, UK. Ethical approval was obtained from the Biomedical & Scientific Research Ethics Committee (BSREC) of Faculties of Medicine and Science at the University of Warwick (REGO-2018-2230).

### Overall study design

We conducted a mixed-methods, repeated cross-sectional, four-arm ("group"), cluster randomised controlled trial to evaluate the effects of monetary incentives to stimulate organisational efforts to improve employee health and wellbeing. Quantitative outcome assessors were blinded to group allocations. There were two intervention groups (Groups 1 and 2)

comprising different "doses" of the monetary incentive intervention, and two control arms (Groups 3 and 4) that received no monetary incentive. The second control arm (Group 4) was used to assess 'reactivity' [14] effects (see 'control conditions' below).

## Deviation from study protocol

There were two noteworthy deviations from the plan for the quantative outcomes data collection outlined out in the study protocol [15]. Each of these occurred towards the end of endline research (Dec 2019–Jan 2020), were approved by the ethics committee and the Principal Investigator, and were made in order to prevent attrition.

First, the mode of interviews for quantative data collection conducted with employees and employers was not limited to face-to-face interviews (as had been planned). Telephone interviews were also conducted with staff, whereas initially, only telephone qualitative interviews had been planned. This was due to many staff being difficult to reach in person due to factors such as remote working, travelling, etc. In total, 473 employee and 67 employer telephone impact interviews were conducted. Some SMEs were offered a web survey that contained a link to the ethics form and the primary outcome measure ('Does your organisation take positive action on health and wellbeing?') only. This change was made in order to accommodate SMEs who had insufficient time to complete the full research process, and to minimise attrition. At endline, 60 employee and six employer web surveys were filled out.

Second, some data collection researchers assisted the WMCA in contacting staff within SMEs to obtain consent for interview. This occurred because many SMEs at endline were not responsive to contacts from the WMCA, therefore researchers supported the WMCA to increase capacity. Any researchers who assisted the WMCA in seeking and/or obtaining consent from an SME did not subsequently conduct interviews with staff in that SME.

Additionally, SMEs that changed in size during the evaluation (to have less than ten employees or greater than 250) were not excluded if the structure and ownership of the organisation remained the same. At baseline, those who collected qualitative data did not collect quantitative data from the same SMEs; however, at endline, some quantitative and qualitative interviews were conducted within SMEs by the same researcher. The qualitative interview always followed the quantitative interview. This was conducted in order to ease scheduling under tight timelines.

Finally, some analyses were conducted that were additional to those outlined in the protocol [15].

## Participants

**Clusters.**   Clusters were small and medium-sized enterprises (SMEs–organisations with 10 to 250 staff) located in the West Midlands.

**SME eligibility.**   SMEs were eligible if they were: 1) located in the West Midlands Combined Authority; 2) receptive to implementing health and wellbeing behaviour change within workplaces; 3) willing to provide organisational level data; 4) willing to allow time for employees and senior executives to be interviewed; 5) registered with Companies House. SMEs were recruited through a planned and targeted emailing and social media campaign via key public and private sector leaders and colleagues [15].

**Employee eligibility.**   Employees were eligible if they were: 1) 16 years of age or older; 2) on an employment contract with the SME, and 3) willing to provide written consent.

**Employer eligibility.**   Employers were volunteers with senior-level responsibilities identified by each SME in conversation with the Local Government Implementation Team (WMCA).

### Randomisation and masking

**Comparison groups.**   SMEs were randomly allocated by an independent statistician using a computer-generated sequence to one of four groups (arms) in a 1:1:1:1 ratio:

- Group 1 (high incentive—intervention): 100% of the monetary incentive

- Group 2 (low incentive—intervention): 50% of the monetary incentive

- Group 3 ("single control"): no monetary incentive with baseline, midline and endline observations

- Group 4 ("double control") no monetary incentive with endline observations only to enable us to examine 'reactivity' [14] effects (see 'control conditions' below)

There are multiple pairwise and group comparisons that can be estimated from multi-arm trials [16]. Here, we report estimates of the following planned [15] "pairwise" comparisons:

1. High incentive vs no incentive;

2. Low incentive vs no incentive;

3. High vs low incentive (for dose effect).

We also report on 'reactivity' [17] effects (see 'control conditions' below) and temporal change (from baseline to endline).

**Randomisation.**   To allocate SMEs to trial arms, we used covariate-constrained randomisation [18] on the basis of number of employees and SME industry type according to the UK Standard Industrial Classification of Economic Activities 2007 (SIC): 1) manual and secondary sector, 2) service and tertiary sector, 3) social and public sector (see protocol for further details [15]). Within each SME, we randomly selected up to 15 employees from lists of employees provided by the SME. This selection was completely random with no constraint.

**Masking.**   Recruitment, enrolment, and delivery of intervention were carried out by the Local Government Implementation Team. Outcome assessors were blinded to allocation [15]. The statistician had no role in delivering the intervention or data collection and was blinded to SME and employee identity.

### Controls and interventions

**All conditions (usual care).**   All four groups were offered a free set of documents (information pack) providing support to organisations on improving the health and wellbeing of their workforce (https://www.wmca.org.uk/what-we-do/thrive/thrive-at-work/). This pack was available to any organisation that visited the website.

**Network meetings.**   Groups 1–3 were provided with facilitated 'network meetings'. The idea of these meetings was to help the organisations make best use of the incentive by providing structured opportunities to form 'communities of practice', share experiences, and learn from each other. Such an approach is consistent with the principles of 'network organisational architecture' [19]. These network meetings were open to members of a specific trial group with no cross-membership and observed by qualitative researchers. They are described in more detail in S1 Text.

**Control conditions.**   Two control conditions (Groups 3 and 4) received no monetary incentive. The Work and Health Unit, which funded this study, wanted to be able to assess the effect of (relatively expensive) financial incentives net of (relatively inexpensive) network meetings. This required a control group that also had access to network meetings–Group 3. However, we noted that Group 3 received considerable additional attention compared to a

counterfactual 'care as usual. Group. First, Group 3 was exposed to multiple (qualitative and quantitative) observations that might cause a type of Hawthorne effect [14]. Second, Group 3 was exposed to the network meetings described above. We did not have resources to tease apart these two putative effects, but we were able to implement a further control (Group 4) that enabled us to measure any combined effect of 'reactivity' to multiple observations and/or to network meetings. Group 4 was thus excluded from network meetings and all but endline observations. We were therefore able to estimate reactivity by comparing Group 3 (which was exposed to network meetings and multiple observations) with Group 4 (which received no attention apart from randomisation prior to group allocation).

**Intervention.**   The monetary intervention is described following the Template for Intervention Description and Replication Guide [20]. SMEs allocated to Groups 1 and 2 received two bank transfers from the Local Government Implementation Team.

The first transfer was not conditional on performance and depended only on group allocation (high or low incentive) and number of staff. SMEs in Group 1 received £60 per employee, while SMEs in Group 2 received £30 per employee.

The second instalment was conditional on demonstrating progress towards achieving up to 43 criteria, where each criterion reflected progress improving employer 'offer' (services to improve health and wellbeing). Organisations were required to complete an on-line self-assessment form (Table A in S1 Text). Group 1 SMEs meeting all 43 criteria could receive up to an additional £140 per employee in the second transfer, and SMEs in Group 2 an additional £70 per employee. However, payments were capped at £10,000 for SMEs in Group 1 and £5,000 in Group 2 (see Table 1). Not only were there many different criteria, but payment for compliance with a given criterion varied according to how difficult it was to achieve; for example, some criteria required more senior staff input than others. Payment was graded according to a scale of difficulty as shown in Table B in S1 Text. Following submission of the self-assessment form, a member of the Local Government Implementation Team visited the organisation to confirm or amend the organisation's performance assessment. The assessment process on

**Table 1. Description of the intervention.**

| **Magnitude of first and second incentive payments** | |
| --- | --- |
| Group 1 (100% of the incentive) | eligible to receive up to £200 total per employee—£60 first unconditional payment, and up to £140 second conditional payment |
| Group 2 (50% of the incentive) | eligible to receive up to £100 total per employee—£30 for first unconditional payment, and up to £70 second conditional payment |
| **Payment caps** | |
| Group 1 (100% of the incentive) | maximum of £10,000 (or 50 x £200 per employee) |
| Group 2 (50% of the incentive) | maximum of £5,000 (or 50 x £100 per employee) |
| **Incentive instalment** | |
| First instalment of 30% | unconditional–following baseline assessments |
| Second instalment up to 70% | conditional–completion of criteria |
| **Mode of delivery** | |
| Group 1 (100% of the incentive) | bank transfer from Local Government Implementation Team |
| Group 2 (50% of the incentive) | |

Recruitment started in July 2018, and the trial ended in January 2020 (timelines in Fig 1).

which the incentive was calculated was independent of the evaluation described here in line with principles of independent intra-mural and extra-mural evaluations [21].

**Outcomes.** *Quantitative outcomes.* The methods and questionnaires used are described in the published protocol [15]. Briefly, we aimed to capture selected proximal and distal effects of the intervention across a hypothesised causal chain (Fig A in S1 Text); the incentive was designed to improve the employer offer (information, activities, or services about health and wellbeing), which employees would perceive, and employee behaviour would change as a result. Ultimately, this would be reflected in employee health and wellbeing.

The precise wording of the questions for both employees and employers is given in Table C in S1 Text. The topics covered were:

1. Perceptions of the organisation's actions towards promoting health and wellbeing (Outcome 1, the primary outcome), which was based on the NHS Commissioning for Quality and Innovation (CQUIN) question of the annual staff survey [22].

2. Perceptions of the employer offer across each of three specific domains: mental health, musculoskeletal health (MSK), and lifestyle health. For each domain, we asked two questions. The first concerned provision of information regarding health (Outcome 2). The second was about provision of health-related activities or services (Outcome 3).

3. Employee behaviour change across each of the three specific domains: making a conscious personal effort to improve health, taking part in activities at work, and taking part in activities outside work (Outcomes 4 to 6).

4. Employee subjective wellbeing, covering four domains: overall life satisfaction, feelings of worthwhileness of day-to-day activities in life, and experience of the day prior in terms of happiness and anxiety. These outcomes were measured on ordinal scales (0–10) using the four main questions used by the Office for National Statistics to monitor national wellbeing (Outcome 7) [15].

Where appropriate, similar questions were asked of employers (Table C in S1 Text).

**Qualitative data.** We observed 'fidelity' according to whether organisations received the first incentive payment, were eligible for the second payment and, if they were eligible, whether they received it. We asked employers how they used the incentives and explored their motivations and the barriers they faced. We also explored employees' perceptions regarding employer actions and attitudes, what their intentions were, and what barriers they faced. Methods used included document review, observations of the network meetings, and interviews with the Local Government Implementation Team, employers, and employees. All interviews were semi-structured and followed standardised topic guides (available upon request). The interviews were audio-recorded with interviewees' permission, and interview transcripts and notes were prepared based on the recordings.

**Power and sample size.** The sample size was based on a Bayesian "assurance analysis" to determine the probability of estimating an effect size within a given degree of certainty [15]. We designed the trial to achieve an 80% probability of observing a 95% posterior credible interval that excluded zero (or relative risk of one) for dichotomous outcomes, with a baseline of 10% to 50%. For this design analysis we specified a prior for the intraclass correlation coefficient uniform between 0.001 and 0.1, to accommodate both very small and large levels of clustering, and a uniform prior between 0.5 and 0.95 for the cluster autocorrelation coefficient, such that the probability is averaged over plausible values for the clustering parameters. We modelled the intervention effects for the dichotomous outcomes in the low (50%) and high (100%) incentive groups as odds ratios of 1.3 ± 0.1 and 1.5 ± 0.1, respectively [15]. The

probabilities were estimated for different sample sizes using a Monte Carlo simulation-based approach, and fitting the pre-specified model. On this basis, we set out to include 132 SMEs (33 per trial group) and interview a random sample of up to a maximum of 15 employees per SME. We aimed to stop recruitment at 152 SMEs to account for 15% attrition. Further details are published elsewhere [15].

### Analysis

Bayesian hierarchical model-based analyses were conducted. Binomial-logistic regression models were used for dichotomous outcomes and linear models for continuous outcomes. All models included four dichotomous variables indicating if: (1) the cluster had received a high incentive; (2) the cluster had received the low incentive; (3) the cluster was in the "double control"; and (4) whether it was an endline observation. All models were also adjusted for the covariates used in the covariate-constrained randomisation, [15] and cluster and cluster-time random effects. We calculated relative risks and absolute intervention effects from these models using the predicted probabilities under appropriate comparator conditions for the whole dataset. We report the probability the effect was greater than one for relative risks, or zero for absolute effects. We used weakly informative prior distributions for the model parameters [15, 23].

Interview transcripts and notes were analysed, aided by Nvivo. Thematic analysis was used to identify recurring themes in the qualitative data. A coding framework (Table D in S1 Text) was developed based on the interview guide and the logic model [15]. The coded data were used to describe the range of attitudes and perceptions that influenced employee and employer behaviour and to shed light on how the intervention worked in practice.

### Patient and public involvement

Four employers provided feedback on the design of the trial, including the feasibility of randomisation, number of questionnaires, and willingness to provide data. A lay summary of the research was written for dissemination.

### Dissemination plans

The scientific paper and the report will be published on Rand Europe website, West Midlands Combined Authority website, The Work and Health Unit, and where possible a lay summary to SMEs that participated in the trial.

### Role of the funding source

The trial was funded by the UK Government's Work and Health Unit (WHU) supported by the National Institute for Health Research (NIHR) Applied Research Collaboration West Midlands. The WHU funded and co-designed the intervention (with West Midlands Combined Authority) and agreed on the protocol but had no role in data collection, analysis, or interpretation.

(Trial registration: AEARCTR-0003420).

## Results

### Summary statistics

Between July and September 2018, 152 SMEs were recruited and included in the trial. The 152 clusters were randomised to Groups 1–4 (38 clusters per group). SMEs were informed of their allocation after baseline quantitative data had been collected. Twenty-nine (19%) SMEs dropped out before baseline data collection (see Fig 1). Baseline data collection was carried out

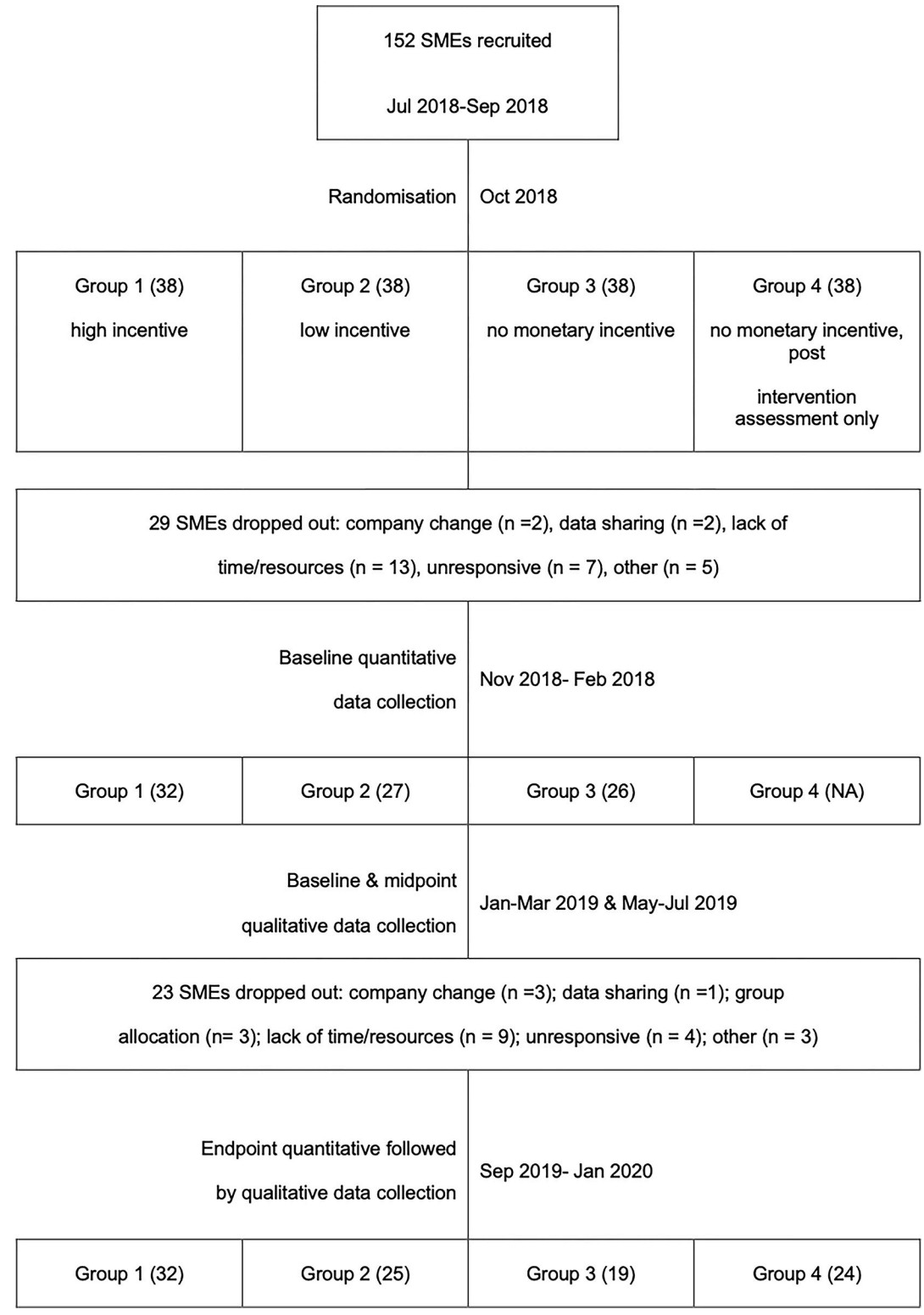

**Fig 1. Cluster flow through stages of the evaluation.** Figure uploaded separately.

**Table 2. SME characteristics at baseline and endline.**

| | Enrolled Group | | | |
|---|---|---|---|---|
| | **1** | **2** | **3** | **4** |
| N (randomised) | 38 | 38 | 38 | 38 |
| Average size | 61.3 | 61.8 | 67.4 | 56.4 |
| Class A (Manufacturing, construction etc.) | 21 | 24 | 18 | 21 |
| Type B (Services) | 45 | 39 | 37 | 42 |
| Type C (Arts, schools, etc) | 34 | 37 | 45 | 37 |
| **Baseline Measures** | | | | |
| N (baseline) | 32 | 27 | 26 | N/A |
| Average size | 65.0 | 53.4 | 67.6 | N/A |
| Class A (Manufacturing, construction etc.) | 19 | 19 | 23 | N/A |
| Type B (Services) | 53 | 44 | 31 | N/A |
| Type C (Arts, schools, etc) | 28 | 37 | 46 | N/A |
| **Endline Measures** | | | | |
| N (endline) | 32 | 25 | 19 | 24 |
| Average size | 65.0 | 49.6 | 79.6 | 49.7 |
| Class A (Manufacturing, construction etc.) | 19 | 20 | 17 | 17 |
| Type B (Services) | 53 | 44 | 33 | 33 |
| Type C (Arts, schools, etc) | 28 | 36 | 50 | 50 |

in the 85 SMEs remaining in Groups 1 to 3, and endline data collection was carried out in 100 SMEs across all four groups.

A summary of the cluster-level covariates by trial at enrolment, baseline, and endline stages of the evaluation is shown in Table 2. There were some differences in these covariates between groups at randomisation: mean SME employee numbers ranged from 56.4 to 67.4 between groups. Employee characteristics by trial group at the baseline and endline stages of the evaluation are shown in Table F in S1 Text. In total, there were 720 employee interviews analysed at baseline (mean per SME = 8.5) and 867 at endline (mean = 8.7). Individual-level covariates remained generally balanced between trial arms at each time-point.

## Quantitative comparison of employee outcomes across intervention groups and controls

The results relating to the primary outcome and the dichotomous secondary outcomes (1–6) are presented in Fig B and Table G in S1 Text, where we show the estimated adjusted intervention effects. Results for the secondary linear items (Outcome 7) are in Fig 2. Crude outcomes for each group are reported in Table H in S1 Text, and the reactivity effect and temporal trends are in Table I in S1 Text.

## Employee perception of employer actions

**Primary outcome (Outcome 1, primary outcome, "Positive action").** Both high and low incentive groups were associated with small increases in employees' perceptions of "positive action": 5 percentage points (pp) [95% CrI -3, 21] and 3pp [–9, 17] respectively, corresponding with probabilities that the effect was positive of 91% and 77%, respectively.

**Awareness of information provided by the employer (Outcome 2).** Regarding the high incentive, there were substantial improvements of 17pp [0, 44], 18pp [1, 40] and 26pp [2, 55] across the three domains of mental health, MSK and lifestyle health information provision.

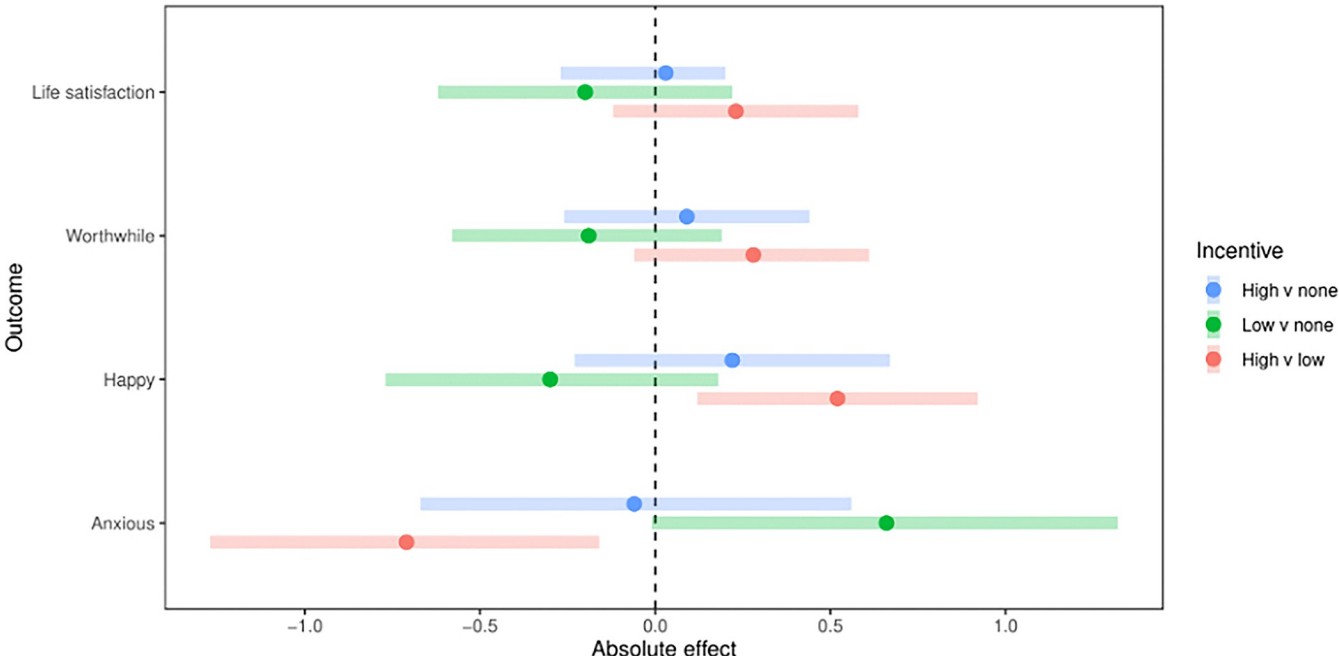

**Fig 2. Absolute intervention effects of incentive on employee subjective wellbeing items note.** Showing results from model-based analyses for high versus no incentive, low versus no incentive, and high versus low incentive. Higher scores reflect better wellbeing except for anxiety, where higher scores reflect worse wellbeing. Figure uploaded sepereatly.

The probabilities that these effects were positive were 98%, 99% and >99% respectively. The lower incentive was associated with smaller effects (3pp [–18, 26], 5pp [–11, 24] and 17pp [0, 44], respectively). The corresponding probabilities of a positive effect were 64, 72 and 98%. The direct comparison of high vs. low incentive was consistent with a dose effect, with probabilities of the higher doses having more of an effect at 97% for mental and MSK health and 90% for lifestyle health.

**Awareness of activities and services provided by the employer (Outcome 3).** Again, the higher incentive was associated with substantial improvements in the proportion of employees who perceived positive change (12pp [–3, 32], 20pp [1, 47], 30pp [3, 58]) across the three domains. The corresponding probabilities that these effects were positive were 94%, 99% and >99% respectively. The lower incentive had smaller effects in the same direction with 5pp [–13, 25], 11pp [–4, 37] and 31pp [3, 61], and respective probabilities of 72%, 90% and >99%. The dose effect was confirmed for mental health and MSK but not lifestyle.

## Employee behaviour change

**Outcomes 4, 5, and 6.** The above changes across perceptions of the offer were not reflected in changes in employees' behaviour. Across all three domains, there was little improvement at either incentive level. This applied to changes in the proposed behaviour (Outcome 4), participation in activities offered by the employer (Outcome 5), and in participation in activities outside the workplace (Outcome 6).

## Employee subjective wellbeing

**Outcome 7; Fig 2 and Table 3.** Changes in reported wellbeing were mixed with no consistent improvements at either level of the incentive (see Table I in S1 Text).

**Table 3. Estimated intervention effects on employee wellbeing items comparing High vs No incentive, Low vs No incentive, High vs Low incentive.**

| Outcome | High versus no incentive | | Low versus no incentive | | High versus low incentive | |
|---|---|---|---|---|---|---|
| | Abs. eff. (95% CrI) | Prob. | Abs. eff. (95% CrI) | Prob. | Abs. eff. (95% CrI) | Prob. |
| Life satisfaction (0–10) | 0.03 (-0.27, 0.2) | 56% | -0.20 (-0.62, 0.22) | 18% | 0.23 (-0.12, 0.58) | 90% |
| Worthwhile (0–10) | 0.09 (-0.26, 0.44) | 69% | -0.19 (-0.58, 0.19) | 17% | 0.28 (-0.06, 0.61) | 95% |
| Happiness (0–10) | 0.22 (-0.23, 0.67) | 83% | -0.30 (-0.77, 0.18) | 11% | **0.52 (0.12, 0.92)** | **99%** |
| Anxiety (0–10) | -0.06 (-0.67, 0.56) | 42% | **0.66 (-0.01, 1.32)** | **97%** | -0.71 (-1.27, -0.16) | 1% |

*Note*. Absolute effects and 95% credible intervals along with the probability (prob.) the effect is greater than zero. Results with "strong evidence" (probability >95%), "fair evidence" (probability 80% - 95%) of an effect are highlighted, and "little evidence" (probability <80%) are not highlighted. Higher scores reflect better wellbeing except for anxiety, where higher scores reflect worse wellbeing.

**Reactivity effects.** There was little to no quantitative evidence of reactivity to baseline or process measurements, or invitations to the observed network meetings, influencing outcomes (Table I in S1 Text).

**Temporal trends.** There were improvements from baseline to endline on employee responses perceptions of employer offerings (Outcomes 1–3), employees' conscious efforts to change their mental and MSK health (Outcome 4), and on the wellbeing items for satisfaction, worthwhile, and happy (Outcome 7).

**Employer interviews.** There were 85 employers interviewed at baseline and 94 at endline (six were unavailable at endline). Responses from employers in the SMEs qualitatively mirrored those of the employees. For example, a greater proportion of employers perceived that they had taken "positive action" at endline than baseline, with the greatest proportion in Group 1 at endline. However, given the lack of random sampling and small sample size (one per SME), we did not estimate intervention effects or provide further inference. Table J in S1 Text reports employer outcomes.

## Qualitative insights

**Fidelity.** All organisations in incentive Groups 1–2 received the incentive payments for which they were eligible. As shown in Table 4, SMEs in Group 1 made more progress towards achieving the second conditional incentive payment. The median number of criteria completed was higher in Group 1, as were the proportion of SMEs that qualified for any amount of the second payment and the proportion of SMEs that qualified for all of the second payment amount. This was consistent with the survey findings that the higher dose of the incentive had a stronger effect.

**Motivation.** Most employers believed that the incentive would positively impact staff health and wellbeing, but most stressed that the incentive had not been their main motivation to sign up for the trial.

**Table 4. SME progress towards achieving second conditional incentive payment.**

| | Group 1 (high incentive) | Group 2 (low incentive) |
|---|---|---|
| No. completed criteria—median (inter-quartile range) | 11.5 (22) | 2 (20) |
| Qualified for any payment (> 0/43 criteria) | 78% (25/32) | 56% (14/25) |
| Qualified for all payment (43/43 criteria) | 21% (7/32) | 4% (1/25) |

*Note*. Progress was based on number of completed criteria.

*'I did not even know about the grant [monetary incentive]–my boss sent me an email with info about some money going through and I didn't know what it was. This was not a factor at all. Genuinely it did not occur to me.'*

**Discretion over funds.**   Employers in Groups 1 and 2 had split views overs the discretion they were given to spend the incentive. Although some appreciated the flexibility, others wished they had more guidance on spending priorities. For example: "We are used to getting money which is actually quite restricted but this one is really relaxed." And: "I would like to have an example from the WMCA of what sort of things we could be looking at that we can spend the grant money [monetary incentive] on!" Some employers reported a lack of understanding of the expected incentive amount to be received and the conditions and requirements tied to the incentive; a likely reflection of the complexity of the payment algorithm.

*'I think it [the monetary incentive] is a very good thing but we would need some assistance in suggestions for how to do things.'*

**Differences across groups.**   The main differences across groups appeared in two areas: incentive spending and attendance at network meetings. SMEs in the high incentive Group 1 spent more on staff time and training, individual-level staff incentives, changes to facilities, and investment in equipment. Staff in Group 1 were also more likely to have attended the network meetings.

**Employers versus employees.**   There was a contrast between employers who described a wide range of policies, procedures and health and wellbeing initiatives in place before the study began versus employees who appeared uninformed of these amenities. Most employers stated that they had implemented further changes during the study period. For example, some high incentive organisations improved the space for employees to relax, others recruited external experts to help develop a wellbeing strategy and some offered drop-in sessions for staff.

*'I want to put plants in all the offices because I was looking at the mental health and wellbeing and it was saying that if you have plants, oxygen, visually. That is a really easy win, isn't it? Simple things, simple wins I call them.'*

**Employer barriers.**   Most employers in the intervention groups reported commitment of resources to support health and wellbeing activities. However, employers reported barriers, such as the pressure of day-to-day business, 'red tape' and logistic difficulties in arranging for staff to participate in activities.

*'We are struggling [to make changes]. We have got everything in place but I cannot demand that this is prioritised over the client work. I cannot because the individuals themselves they know and we don't want that to affect their performance, I would say that as a company we are really supporting it and we really want to be part of this and we are giving time to it but there has to be a balance. And sometimes, this sort of stuff, when you try to collate the information it is time consuming. We have got everything but we need to have everything into place and have them uploaded and that is going to take some time.'*

**Employee barriers.**   While many employees interviewed early in the study said that they were looking forward to health and wellbeing activities provided by their employer, only a small proportion later stated that they had participated in such activities. The reasons for lack of uptake included lack of relevance to their particular needs, lack of time, or no perceived

health need. Most interviewed employees felt that their employer had not identified or responded to their specific health needs, even among organisations that had consulted the workforce.

*'So just even understanding that anything extra we're asked to do is over and above what we're already doing and there needs to be time allowed for that and I sometimes think that the managers in the organisation don't truly appreciate what we're trying to fit in in a day. I'm very happy to do this but there's no way I could have done it on the Monday as suggested, in the middle of a very, very busy day.'*

**Employee participation.**   There were, nevertheless, examples of employee participation in some activities made available in organisations receiving incentives, including a health and wellbeing workshop, health insurance, free flu vaccinations, notice boards, social events, training, walking meetings, yoga, and counselling. Interviewed employees who participated in new activities said they were more aware of the importance of health and wellbeing at work and of mental health issues.

*'I'll be very likely only if it's relevant to help my current state and how I feel. I'd always be open to opportunities that they offer, but for me it'd have to be relevant for my circumstances and situation. So I wouldn't take part in a programme just for the sake of taking part in a programme if I didn't see any benefit for me.'*

## Discussion

### Effects of the incentives on employee perceptions of the employer health 'offer'

Our primary outcome was the answer to the question 'does your organisation take positive action on health and wellbeing?' While the proportion of employees responding affirmatively to this question increased, the effect was small and accompanied by high uncertainty. However, when asked about organisational provision of information regarding specific domains (mental, MSK, and lifestyle health), estimated effects were much larger, and, in the high incentive group, the probability the effect was positive exceeded 97% in each of the three domains. This pattern was broadly repeated when employees were asked whether the organisation provided activities or services to promote health, suggesting that the incentive was causal of a change in employee perceptions of employer behaviour. Other aspects of the results also support such a conclusion, including the employers' accounts, the observed effect of 'dose' (on both 'fidelity' and outcomes), the magnitude of effect sizes, the consistency of effects over multiple categories (Fig B in S1 Text), and the qualitative data.

These broadly positive findings raise two questions; 1) why did employees perceive more change on the three specific domains in the secondary outcomes than on the consolidated primary outcome, and 2) why, if the employees recognised that the employers were making an effort, did they not respond by changing their behaviours?

### The primary outcome vs. other outcomes based on employee perceptions of the employer health 'offer'

Our baseline survey results concerning the dichotomous primary outcome were similar to those reported by NHS staff. In the 2018 NHS staff survey, which contained over 400,000 staff

responses, over 80% of respondents gave a positive response [22]. These high scores raise the possibility that there is relatively little headroom for improvement on this outcome. Moreover, this question might be inherently insensitive because it is vague and general, while questions targeted at particular employer actions (such as mental and MSK health) are more evocative. In short, the general question may be a rather insensitive tool.

## Discordance between employee perceptions of employer actions and employee behaviour

Our findings show that employers made an effort to improve their 'offer' and that employees recognised this effort. This suggests that the incentive was of sufficient size, at least in the high incentive group, and that the complexity of the payment algorithm did not prevent it from having an impact on employer behaviour. However, these positive perceptions did not translate into reciprocal changes in employee behaviour; point estimates were close to no change in all comparisons, even in the high-incentive group. There is a large literature and much debate about the ineffectiveness of many workplace interventions in changing employee behaviour [24]. We focus on two broad reasons for the 'failure in translation' in our context: the organisational capacity to make best use of the incentive and limitations of single studies of limited duration.

## Organisational capacity to make best use of the incentive

An incentive is only as good as the use made of it, and organisations may have lacked the capacity to optimise their offer to the workforce [4]. This idea is supported by those employers who said that they would have appreciated more support in how to spend the money and by employees who said their needs had not been met. While the incentive was accompanied by network meetings [19], this level of support fell well short of the level provided in some recent US trials. One trial of staff working in eight intervention and eight control schools in Los Angeles found that combining participatory committees with school-level stipends and prizes reduced weight gain [25]. Another trial of 20 intervention and 140 control worksites within a single company spread across eastern states found that intensive wellness programming, including individual counselling, team activities (in eight modules over four to ten weeks), and individual (though not organisational) incentives produced a positive effect on self-reported behaviours [4]. However, even with greater wellness programming, it can be a challenge to change behaviour: an Illinois trial of a multi-component, individual-level intervention that included supportive activities like biometric screening and physical fitness classes, which showed generally positive effects on beliefs but, as in our study, not on behaviours [24, 26, 27].

## Limitations of single studies of limited duration

Our observations were carried out over one year, while a previous study showed that a return on investment in a health promotion program only occurred after the third year [28]. Not only may there be non-linear returns to scale over time, but also over place, whereby multiple studies over prolonged periods lead to a tipping point [29]. For instance, individual studies of smoking produced small or null results; over time, however, large reductions in smoking behaviour have been achieved in high-income countries. In the sense that many interventions contribute to real change over long periods in complex systems, our results showing that employers responded sufficiently enough for employees to recognise the change may be taken as encouraging rather than the reverse. The necessary, albeit not sufficient, conditions for behaviour change were achieved.

### Reactivity effects

We were concerned that the additional attention paid to the control group could bias the results. First, they took part in baseline data and qualitative data collection. Second, staff could attend network meetings that were observed by researchers, albeit only with other SMEs in their group. Since study outcomes can be 'reactive' to activities to which both the intervention and control groups are exposed [14], we instituted a 'double control' group that received none of the above three activities before endline data collection using a modified Solomon design [30]. However, the findings did not provide any evidence that outcomes were affected by reactivity effects from observations or network meetings.

### Strengths and limitations

Some of the key strengths of this research are that the results are robust to the influence of reactivity to measurement and meetings, protected from some aspects of detection bias because of interviewer blinding, can be interpreted causally because of the use of random allocation, incorporate baseline as well as endline observations, include employer as well as employee observations, and combine quantitative and qualitative data. Another strength lies in observations along a causal chain that enables us to 'triangulate' findings to provide suggestions about likely mechanisms [31]. Thus, our inferences are based on the totality of the evidence, rather than any individual point estimate or inferential statistic. Given that we have also eschewed a "statistical significance" approach and used Bayesian methods to estimate intervention effects, we have not "corrected" for multiple comparisons.

Limitations include that the results may not generalise to other samples and contexts, although the wide variety in sectors of participating organisations over a wide area provides reassurance on this point; we did not include observer measures such as productivity or absenteeism; and the drop out of SMEs might not have been random. Attrition was lower in the high incentive group. If it were the case that organisations with low interest in employee health and wellbeing were more likely to drop out when there was no incentive, then this would bias the findings against the positive effect observed in the high incentive group. Our follow up was limited such that we might have missed the effects that take longer to accrue or possible demotivating effects of withdrawal of an incentive [32].

### Research in context

**Evidence before this study.**  There is a growing interest in improving population health and wellbeing by intervening in the workplace, which is a setting offering access to large groups of people. The benefits of healthy behaviours acquired at work may spill into daily life and continue into later life. Workplace interventions have been the subject of a large literature, which was compiled in a recent overview [5]. However, very few studies have examined the effects of monetary incentives targeted at organisations despite use of monetary incentives by both the US and UK governments. We examined over 900 studies identified by three of these reviews [6–8], locating only five studies that provided organisational-level monetary incentives alone as an intervention (as opposed to individual-level incentives or those embedded within multi-component interventions) [9–13]. There is a clear lack of high-quality experimental evidence on the effect of organisation-level incentives on the effectiveness of workplace wellbeing and health programmes.

**Added value of this study.**  We conducted a single-blinded cluster RCT to examine the effectiveness of organisational level monetary incentives to encourage small and medium-sized enterprises to take action to improve the health and wellbeing of their employees. We examined a causal chain starting with payment of the incentive, effect on employer actions,

employee perceptions of those actions, and effects on employee behaviour and wellbeing. We find that employers respond and take action due to the incentive, and employees perceive that action has been taken. However, this does not translate into employee behaviour change related to health and wellbeing, or to wellbeing itself. Our results suggest that organisations' actions failed to meet employees' needs and that practical barriers, such as time pressures, prevented uptake. Our study also examined the incentive's size and found "dose-response" effect where larger incentives were more effective. This supports causal inference regarding the effect of the monetary incentive. The consistency of the observed effects over a large range of different employer actions and employee behaviours adds credibility to our findings. Our trial was also novel in this area because we accounted for the potential of study processes themselves to impact trial outcomes independently of the intervention, including interviewing participants about health and wellbeing and the observed meetings about health and wellbeing. However, no evidence of such an effect was identified.

**Implications of the available evidence.** A monetary incentive can change employer behaviour and the workplace environment. A dose effect was observed. However, changing the workplace environment does not necessarily translate into changes in employee behaviour. It is possible that coupling an incentive with greater organisational support in how to change employee behaviour would have a larger effect. In the meantime, our findings show that the necessary but not sufficient conditions for improved health and wellbeing were achieved.

## Supporting information

**S1 Checklist.**
(DOCX)

**S1 Text. Network meetings. Table A.** Self-assessment criteria description. **Table B.** Self-assessment criteria payments. **Fig A.** Effect of Incentive on Employer "Offer"–A Simplified Causal Chain Linking Intervention to Outcomes. **Table C.** Quantitative outcome measures. **Table D.** Process evaluation codes. **Table E.** Positive action–employees' perception of amount of action. **Table F.** Summary of employee-level covariates at baseline and endline (11 months) by group. **Fig B.** Employee Perception of Employer and Employee Behaviour Change Outcomes. **Table G.** Employee Outcomes Model-Based Analyses. **Table H.** Summary of. crude employee-level outcomes by baseline/end line and trial group (%). **Table I.** Model-based estimates of the reactivity effect and endline versus baseline. **Table J.** Summary of employer-level crude outcomes by baseline/endline (%).
(DOCX)

**S1 File.**
(PDF)

## Acknowledgments

The authors would also like to thank:

Andreas Culora, Analyst, RAND Europe: Contributed to report writing.

Rahma Gulaid, Corporate Support Officer, West Midlands Combined Authority: Contributed to intervention delivery and data collection.

Hassan Jamil, Corporate Support Officer, West Midlands Combined Authority: Contributed to intervention delivery

Chunmun Kamal, Corporate Support Officer, West Midlands Combined Authority: Contributed to intervention delivery and data collection.

Sharon Lindop, Accreditation Manager, West Midlands Combined Authority: Contributed to SME communications, data collection, delivery of intervention, data collection, network meetings and Thrive at Work accreditation.

Edward Marson, Corporate Support Officer, West Midlands Combined Authority: Contributed to intervention delivery.

Nadja Koch, Research Assistant, RAND Europe: Contributed to the design of data collection tools, qualitative data collection, observation of network meetings and report writing.

Nimmi Patel, Project Manager, West Midlands Combined Authority: Contributed to SME recruitment and communications.

Jessica Perry, Mental Health Commission Coordinator, West Midlands Combined Authority: Contributed to SME recruitment.

Rob Prideaux, Senior Research Leader, RAND Europe: Contributed to the conception of the evaluation, design of data collection tools and drafting the trial protocol.

Magdalena Skrybant, Patient and Public Involvement and Engagement Lead, University of Birmingham: Contributed by writing and revising lay summary.

Chloe Stacey, Corporate Support Officer, West Midlands Combined Authority: Contributed to SME recruitment and intervention delivery.

Alex Sutherland, Senior Research Leader, RAND Europe: Original Primary Investigator of the RAND Europe work package, contributed to the inception of the evaluation and the trial design.

Katey Tetteh, Corporate Support Officer, West Midlands Combined Authority: Contributed to intervention delivery.

Christian van Stolk, Vice President, RAND Europe: contributed to the inception of the intervention and evaluation, and provided quality assurance on the final report.

## Author Contributions

**Conceptualization:** Lena Al-Khudairy, Yasmin Akram, Joanna Hofman, Madeline Nightingale, Sean Russell, Richard J. Lilford.

**Data curation:** Lena Al-Khudairy, Laura Kudrna, Joanna Hofman, Lailah Alidu, Andrew Rudge, Clare Rawdin, Iman Ghosh, Frances Mason, Chinthana Perera, Jane Wright, Joseph Boachie.

**Formal analysis:** Samuel I. Watson, Laura Kudrna, Lailah Alidu, Iman Ghosh, Chinthana Perera.

**Funding acquisition:** Yasmin Akram, Richard J. Lilford.

**Investigation:** Lena Al-Khudairy, Yasmin Akram, Laura Kudrna, Lailah Alidu, Andrew Rudge, Clare Rawdin, Iman Ghosh, Karla Hemming, Sean Russell.

**Methodology:** Yasmin Akram, Samuel I. Watson, Joanna Hofman, Madeline Nightingale, Lailah Alidu, Chinthana Perera, Ivo Vlaev, Richard J. Lilford.

**Project administration:** Lena Al-Khudairy, Sean Russell.

**Supervision:** Yasmin Akram, Richard J. Lilford.

**Writing – original draft:** Lena Al-Khudairy, Richard J. Lilford.

**Writing – review & editing:** Lena Al-Khudairy, Yasmin Akram, Samuel I. Watson, Laura Kudrna, Joanna Hofman, Madeline Nightingale, Lailah Alidu, Andrew Rudge, Clare Rawdin, Iman Ghosh, Frances Mason, Chinthana Perera, Jane Wright, Joseph Boachie, Karla Hemming, Ivo Vlaev, Sean Russell, Richard J. Lilford.

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
