## [Decision Letter · Decision Letter 0]

22 Jun 2022

PGPH-D-21-00689

Evaluation of an organisational-level monetary incentive to promote the health and wellbeing of workers in small and medium-sized enterprises: a cluster randomised controlled trial

Dear Dr. Al-Khudairy,

Thank you for submitting your manuscript to PLOS Global Public Health. After careful consideration, we feel that it has merit but does not fully meet PLOS Global Public Health’s publication criteria as it currently stands. Therefore, we invite you to submit a revised version of the manuscript that addresses the points raised during the review process.

We look forward to receiving your revised manuscript.

Kind regards,

Behdin Nowrouzi-Kia

Academic Editor

Journal Requirements:

1. Please provide separate figure files in .tif or .eps format and remove any figures embedded in your manuscript file. Please also ensure that all files are under our size limit of 10MB.

2. Thank you for submitting your clinical trial and for providing the name of the registry and the registration number. The information in the registry entry suggests that your trial was registered after patient recruitment began. PLOS strongly encourages authors to register all trials before recruiting the first participant in a study.

a) your reasons for your delay in registering this study (after enrolment of participants started);

b) confirmation that all related trials are registered by stating: “The authors confirm that all ongoing and related trials for this drug/intervention are registered”."

Please see http://journals.plos.org/globalpublichealth/s/submission-guidelines#loc-clinical-trials for our policies on clinical trials.

Additional Editor Comments (if provided):

Reviewers' comments:

Reviewer's Responses to Questions

**Comments to the Author**

1. Does this manuscript meet PLOS Global Public Health’s publication criteria? Is the manuscript technically sound, and do the data support the conclusions? The manuscript must describe methodologically and ethically rigorous research with conclusions that are appropriately drawn based on the data presented.

Reviewer #1: Yes

2. Has the statistical analysis been performed appropriately and rigorously?

Reviewer #1: Yes

3. Have the authors made all data underlying the findings in their manuscript fully available (please refer to the Data Availability Statement at the start of the manuscript PDF file)?

Reviewer #1: Yes

4. Is the manuscript presented in an intelligible fashion and written in standard English?

Reviewer #1: Yes

5. Review Comments to the Author

Reviewer #1: As the study also involving qualitative technique and interviews, I don't see any data cited in the findings. It will be useful if the authors could refer to the data when explaining the qualitative data in the results section.

In addition, the authors wrote about thematic analysis for the qualitative data analysis. However, I don't find any example of codes that being generated by this analysis technique. Sample of codes and how it was build into themes would be better for the paper

6. PLOS authors have the option to publish the peer review history of their article (what does this mean?). If published, this will include your full peer review and any attached files.

**Do you want your identity to be public for this peer review?** For information about this choice, including consent withdrawal, please see our Privacy Policy.

Reviewer #1: No

---

## [Decision Letter · Decision Letter 1]

6 Apr 2023

PGPH-D-21-00689R1

Evaluation of an organisational-level monetary incentive to promote the health and wellbeing of workers in small and medium-sized enterprises: a cluster randomised controlled trial

Dear Dr. Al-Khudairy,

Thank you for submitting your manuscript to PLOS Global Public Health. After careful consideration, we feel that it has merit but does not fully meet PLOS Global Public Health’s publication criteria as it currently stands. Therefore, we invite you to submit a revised version of the manuscript that addresses the points raised during the review process.

EDITOR: Please insert comments here and delete this placeholder text when finished. Be sure to:

Indicate which changes you require for acceptance versus which changes you recommendAddress any conflicts between the reviews so that it's clear which advice the authors should followProvide specific feedback from your evaluation of the manuscript

Please ensure that your decision is justified on PLOS Global Public Health’s publication criteria and not, for example, on novelty or perceived impact.

We look forward to receiving your revised manuscript.

Kind regards,

Justice Nonvignon, PhD

Section Editor

Journal Requirements:

Additional Editor Comments (if provided):

Authors are encouraged to address the comments of the reviewer, particularly around clustering

Reviewers' comments:

Reviewer's Responses to Questions

**Comments to the Author**

1. If the authors have adequately addressed your comments raised in a previous round of review and you feel that this manuscript is now acceptable for publication, you may indicate that here to bypass the “Comments to the Author” section, enter your conflict of interest statement in the “Confidential to Editor” section, and submit your "Accept" recommendation.

Reviewer #2: (No Response)

2. Does this manuscript meet PLOS Global Public Health’s publication criteria? Is the manuscript technically sound, and do the data support the conclusions? The manuscript must describe methodologically and ethically rigorous research with conclusions that are appropriately drawn based on the data presented.

Reviewer #2: Yes

3. Has the statistical analysis been performed appropriately and rigorously?

Reviewer #2: No

4. Have the authors made all data underlying the findings in their manuscript fully available (please refer to the Data Availability Statement at the start of the manuscript PDF file)?

Reviewer #2: Yes

5. Is the manuscript presented in an intelligible fashion and written in standard English?

Reviewer #2: (No Response)

6. Review Comments to the Author

Reviewer #2: Minor concern

Abstract

Please check the title: used a mixed-methods cluster randomised trial instead of “cluster randomized trial”

Please define reactivity effect

The impact estimates seem not be statistically significant looking at the 95% Credible Interval. In other words, the intervention seems not to be working especially on employee health behavior or wellbeing outcomes. Please check and confirm both the point and interval estimate. Would you recommend the intervention for larger scale-up based on the impact estimate?,

Check line 120, 125 and 126 for spelling errors

Major concern

Authors must demonstrate in this paper how they arrived at appropriate clustering effect. What was the exact estimate of the design/clustering effect. How did the authors adjust for the multiple arm effect in the estimation of the sample size? Authors must explain how they arrive at 15 employees per SME.

The study accounted for 15% attrition, but the analysis showed that 19% of the SMEs actually attritted. Authors should explain how it affected the overall results and the limitation thereof.

It seems the 95% credible intervals from the Bayesian models looks incredibly large and highly imprecise. It is because of the sample size?. May be a sensitivity analysis using a pure frequentist approach for the respective Hierarchical Bayesian methodology and putting the results side-by-side would be greatly beneficial in this case. This needs to be checked.

A difference in differences analysis could help address the effect of time invariant and varying covariates between the intervention and control SMEs especially when there were some covariate imbalance at baseline

Please specify the empirical model used in the estimation of the model parameters

7. PLOS authors have the option to publish the peer review history of their article (what does this mean?). If published, this will include your full peer review and any attached files.

**Do you want your identity to be public for this peer review?** For information about this choice, including consent withdrawal, please see our Privacy Policy.

Reviewer #2: No

---

## [Editor Report · Decision Letter 2]

26 May 2023

Evaluation of an organisational-level monetary incentive to promote the health and wellbeing of workers in small and medium-sized enterprises: a mixed-methods cluster randomised controlled trial

PGPH-D-21-00689R2

Dear Dr Al-Khudairy,

We are pleased to inform you that your manuscript 'Evaluation of an organisational-level monetary incentive to promote the health and wellbeing of workers in small and medium-sized enterprises: a mixed-methods cluster randomised controlled trial' has been provisionally accepted for publication in PLOS Global Public Health.

Best regards,

Justice Nonvignon, PhD

Section Editor

Dear Corresponding Author,

It is interesting that some of your explanation actually aligns with the reviewer comments (e.g. the lack of impact on your main outcome, employee well-being), yet you make the case that the reviewer lacks understanding of the statistical methods used. It is ok to disagree with a reviewer's comment without necessarily making a judgment about the reviewer's understanding or lack thereof, and an editor will assess that independently. 

With best wishes